# One-Step, Low-Cost, Operator-Friendly, and Scalable Procedure to Synthetize Highly Pure *N*-(4-ethoxyphenyl)-retinamide in Quantitative Yield without Purification Work-Up

**DOI:** 10.3390/molecules27113632

**Published:** 2022-06-06

**Authors:** Silvana Alfei, Guendalina Zuccari

**Affiliations:** Department of Pharmacy (DIFAR), University of Genoa, Viale Cembrano, 16148 Genoa, Italy

**Keywords:** *N*-(4-hydroxyphenyl)-retinamide, *N*-(4-ethoxyphenyl)-retinamide, one-step operator-friendly synthetic procedure, complete characterization, quantitative yield, high level of purity

## Abstract

It is widely reported that *N*-(4-hydroxyphenyl)-retinamide or fenretinide (4-HPR), which is a synthetic amide of all-trans-retinoic acid (ATRA), inhibits in vitro several types of tumors, including cancer cell lines resistant to ATRA, at 1–10 µM concentrations. Additionally, studies in rats and mice have confirmed the potent anticancer effects of 4-HPR, without evidencing hemolytic toxicity, thus demonstrating its suitability for the development of a new chemo-preventive agent. To this end, the accurate determination of 4-HPR levels in tissues is essential for its pre-clinical training, and for the correct determination of 4-HPR and its metabolites by chromatography, *N*-(4-ethoxyphenyl)-retinamide (4-EPR) has been suggested as an indispensable internal standard. Unfortunately, only a consultable old patent reports the synthesis of 4-EPR, starting from dangerous and high-cost reagents and using long and tedious purification procedures. To the best of our knowledge, no article existed so far describing the specific synthesis of 4-EPR. Only two vendors worldwide supply 4-ERP, and its characterization was incomplete. Here, a scalable, operator-friendly, and one-step procedure to synthetize highly pure 4-EPR without purification work-up and in quantitative yield is reported. Additionally, a complete characterization of 4-EPR using all possible analytical techniques has been provided.

## 1. Introduction

The *N*-(4-hydroxyphenyl)-retinamide or fenretinide (4-HPR) (Figure 1) is a synthetic amide of all-trans-retinoic acid (ATRA) first produced in the late 1960s. 4-HPR has been reported to inhibit in vitro several types of tumors at 1–10 µM concentrations [1,2], including cell lines resistant to ATRA and cis-retinoic acid [3,4,5,6,7].

Once established the capability of 4-HPR to prevent the formation of several forms of carcinoma in rat and mouse [1], it is being studied as a chemo preventive agent [8], showing no hematological toxicity in clinical trials [2,8]. Although the mechanism for the antitumor effect of 4-HPR remains to be fully understood and different mechanisms have been assumed [9,10,11,12,13], pre-clinical in vitro studies employing ceramide modulators in combination with 4-HPR suggested that high 4-HPR levels will need to be achieved in the tumor tissue for an optimal anti-tumor effect [11]. Therefore, methods to accurately determine 4-HPR levels in tissues are essential for the pre-clinical development of 4-HPR. Several methods employing HPLC to measure levels of retinoids in tissues have been reported [8,14,15,16,17,18,19,20,21,22,23,24,25,26,27,28,29,30,31,32,33], but easy and reproducible HPLC and LC–MS/MS methods for separating and quantifying 4-HPR and its metabolite *N*-(4-methoxyphenyl)-retinamide (4-MPR) in various tissues have been reported by Vratilova et al. [34] and by Cho and colleagues [35]. Both protocols suggest using 4-EPR as internal standard. In this regard, just three old and incomplete laborious patents [36,37,38] using harmful reactants exist concerning its synthesis (Appendix A), but only one, reporting a synthetic procedure on a scale lower than 50 mg to obtain 4-EPR in modest yield, and after a column chromatography work-up for purification, is fully consultable (Appendix A) [36]. On the contrary, no article is available reporting a detailed procedure to obtain specifically 4-EPR. Particularly, only two works exist, reporting a general procedure to prepare several retinamides, including 4-EPR (Appendix A). However, curiously, even if the synthesis of other amide derivatives of ATRA have been precisely described, that of 4-EPR has not, and no detail has been provided regarding its specific purification, the yields of reaction, and data concerning its full characterization are missing [39,40]. Additionally, only two vendors in the world supply 4-EPR [41,42], thus making it particularly complex to be able to have 4-EPR available before the present work. Indeed, here, a detailed one-step, easy, and low-cost synthetic method to obtain 4-EPR in quantitative yield, with high level of purity, without need of any purification procedure, and without using reagents toxic for humans and environment, has been reported. Additionally, the prepared 4-EPR was fully characterized by determining its melting point range and elemental analysis, by running HPLC and GC-MS analyses, as well as by UV–Vis, FTIR, and NMR spectrophotometry.

## 2. Results and Discussion

Figure 1 reports the synthetic procedure carried out by us to prepare 4-EPR. To be more observable, numbers were included on the chemical structure of 4-EPR to clarify the assignation reported in the peaks list of ^1^H and ^13^C NMR, reported in the Experimental Section.

Particularly, TEA was used to free 4-ethoxyaniline (4-EA) from its hydrochloride salt (4-EAH), DMAP was the catalyst and EDCI the coupling agent. Among the other available carbodiimide derivatives, we retained EDCI—the better choice to simplify the purification work-up of 4-EPR, thus rendering the method easy, fast, and operator-friendly. In fact, while using other coupling agents such as *N,N′*-dicyclohexylcarbodiimide (DCC), tedious chromatographic columns, requiring high volumes of solvents would be necessary, using the basic EDCI, simple acid washings would allow to eliminate the acyl ureic derivative deriving by the rearrangement of the adduct which forms between ATRA and EDCI [43] which is known to be the main side product affecting this kind of reactions [43]. Additionally, the acid washing allowed to remove simultaneously also the unreacted EDCI and the catalyst (DMAP). The unreacted ATRA used in excess was instead removed by subsequent basic washings. The pureness of the prepared 4-EPR was first investigated by thin layer chromatography (TLC), using *n*-hexane/acetone (6/4) as eluent, evidencing that the simple and fast purification procedure carried out by us was successful. In fact, the TLC profile (Rf = 0.61) showed absence of any side product and of unreacted reagents (Figure 2).

The pureness of 4-EPR (100%) was confirmed by HPLC analysis (Figure 3). 

Furthermore, the narrow range of the melting point (178−179 °C), and elemental analysis further confirmed its purity (see Experimental Section). The characterization of 4-EPR was completed acquiring the UV–Vis, FTIR, ^1^H, ^13^C NMR, and GC-MS spectra. The obtained data are reported in the Experimental Section, while the spectra are observable in the following Figure 4 and Figure 5b (UV–Vis and FTIR spectra), and in Appendix A (^1^H, ^13^C NMR, and GC-MS spectra), in Appendix A.

Particularly, in the FTIR spectrum of 4-EPR (Figure 5b), bands typical of the amide group are clearly visible at 1638 cm^−1^ (C=ONH) and at 3293 cm^−1^ (NH), while the band at 1688 cm^−1^ belonging to the carboxylic group of ATRA (Figure 5a) is no longer detectable.

Importantly, in a first attempt to prepare 4-EPR following the generic procedure and stoichiometry proposed by Campos-Sandoval et al. to obtain several amide derivatives of ATRA omitting the specific description and results concerning the obtainment of 4-EPR [40], we obtained 4-EPR in very low yield (8.3%). Additionally, we obtained the little 4-HPR polluted by a side product (6.5%) that was isolated as a brown oil and was identified as the bis (retinoic acid) anhydride (BRAA) by FTIR analysis. Particularly, the FTIR spectrum of the unknown side product unequivocally evidenced the presence at 1741 cm^−1^ of the stretching band due to the strongly conjugated OC=OO group typical of the anhydrides (Appendix A. We are confident that the order of the addition of reagents is essential to limit BRAA formation. In this regard, mixing first ATRA and EDCI and adding the phenylamine later, as suggested by Campos-Sandoval et al. [40], could have favored the formation of BRAA, since in the absence of the amine, the ATRA itself can react with ATRA activated by EDCI to form the anhydride (BRAA). Consequently, in the procedure herein proposed by us, we have changed such order, managing to totally avoid the formation of BRAA. As abovementioned, in the work by Campos-Sandoval et al., details concerning the preparation of 4-EPR are missing, so we cannot compare the first results obtained by us following their procedure with those obtained by the authors previously. Anyway, the very low yields of both 4-EPR and the side product evidenced the presence of many unreacted material, which probably was due to both an incorrect stoichiometry and to the absence of a catalyst. Particularly, we inverted the proposed stoichiometry, using an excess (1.2 equivalents) of ATRA in place of that of 4-EAH, which was the limiting reagent. Additionally, we used 0.6 equivalent of DMAP as catalyst and a DCM/DMF mixture as solvent since DCM alone did not dissolve neither ATRA nor 4-EAH. By the procedure proposed by us, we enhanced the reaction yield by 12 times (R 99.9% vs. 8.3%), obtaining a product with high degree of pureness (100% by HPLC), without further purification and no side products. Similarly, also the other work found in literature [39], which use a different procedure, like that of the patent by Curley et al. [36], does not report any detail concerning the specific synthesis of 4-EPR, the yield of reaction, the purification procedure, and any analytical data. HPLC analysis for determining the Rf of 4-EPR and confirming its purity, as well as FTIR and elemental analysis were not performed by Curley et al. [36], hence this study is the first one reporting them. Other characterizations, such as UV–Vis, NMR analyses, and melting point were reported by Curley [36], while only the melting point was described in the patent by Maryanoff [37]. Curiously, while our UV–Vis and NMR spectra were compatible with those by Curley et al., the melting point range stated in the two patents are both very different between each other (158–159 °C in the first and 188–189 °C in the latter) and different from that determined by us (178–179 °C). A reaction yield of 62.8% has been described by Curley et al. starting from a very low amount of ATRA (44.8 mg), thus establishing that by the procedure developed by us, we improved the yield of 4-EPR by 1.6 times, working on a scale 7.5-fold higher. Additionally, the procedure proposed by Curley involves a long timing and laborious solid phase synthesis using high-cost resins, and a chromatographic column to purify 4-EPR, using considerable amounts of organic solvents depending on the scale of the reaction. Consequently, it is not suitable for scaling up as the procedure herein reported by us. Furthermore, the procedure suggested by Curley et al., as well as those reported in the other patents [37,38], need to convert ATRA in its acyl chloride, before the coupling with phenetidine, thus involving an additional step compared to that developed by us, and consequently, extending the times of the synthesis. Another advantage of the synthesis herein developed consists of having avoided the use of highly hazardous reagents such as esachloroacetone which is harmful if swallowed, and toxic to aquatic life with long lasting effects [44], of tetrafluoroftalic anhydride which causes skin corrosion/irritation, and damage to eye and respiratory tract [45], as well as the use of highly irritating oxalyl chloride [46] or SOCl_2_ [47].

## 3. Materials and Methods

### 3.1. Chemicals and Instruments

All reagent and solvents were of analytical grade and were purchased by Merk—Italy—Sigma-Aldrich (Milan, Italy). Solvents were purified by standard procedures, whereas reagents were employed as such, without further purification. The melting range of 4-EPR was determined on a 360 D melting point device, resolution 0.1 °C (MICROTECH S.R.L., Pozzuoli, Naples, Italy). Elemental analyses were performed on an EA1110 Elemental Analyser (Fison Instruments Ltd., Farnborough, Hampshire, England). Organic solutions were evaporated using a rotatory evaporator Rotavapor^®^ R-3000 (Buchi, Cornaredo, Milan, Italy) operating at a reduced pressure of about 10–20 mmHg. 

### 3.2. Procedure for the Preparation and Isolation of N-(4-ethoxyphenyl)-retinamide (4-EPR)

ATRA (335.3 mg, 1.12 mmol) was dissolved in 6 mL of DCM and complete dissolution was obtained upon the addition of 2 mL of DMF. In the meantime, in a tailed test tube equipped with a magnetic stirrer and carefully flamed in a nitrogen stream, 4-EAH (161.5 mg, 0.9301 mmol) was mixed with TEA (94.1 mg, 0.9301 mmol, 129 µL), DMAP (68.1 mg, 0.5581 mmol) and EDCI (214.0, 1.12 mmol) in DCM (1 mL) and DMF (1.5 mL), obtaining a fine suspension. To this suspension the solution of ATRA was added and the obtained reaction mixture was stirred at room temperature for about 24 h and then added with ethyl acetate (EtOAc, 5 mL). The organic phase was washed with 10% KHSO_4_ (3 × 10 mL), then with NaOH 15% (3 × 10 mL), and finally with H_2_O (3 × 10 mL), to neutral pH. The organic phase was then treated with anhydrous MgSO_4_ and left overnight. Upon removal of solvents at reduced pressure and at temperature not exceeding 60 °C, 4-EPR was obtained as dark yellow crystals (390.2 mg, 0.9300 mmol, 99.99% yield), which were stored at −20 °C protected by light.

*N-(4-ethoxyphenyl)-retinamide (4-EPR).* Isolated yield 99.9%. Melting point 178−179 °C. Purity 100% by HPLC. UV–Vis (MeOH/acetonitrile 1/1): λ _max_ = 361 nm (ε = 52,030). FTIR (ν, cm^−1^): 3293 (stretching N-H); 3025 (stretching = C–H); 2973, 2922, 2861 (methyl and methylene C–H stretching); 1638 (amide C=O stretching); 1051 (C–O stretching).

^1^H NMR (400 MHz, CD_3_OD): 1.16 (s, 6H, 16- and 17-CH_3_), 1.23 (t, 3H, -OCH_2_CH_3_, *J* = 7.0 Hz), 1.34 (m, 2H, 2-H), 1.55 (m, 2H, 3-H), 1.72 (s, 3H, 18-CH_3_), 1.98 (s, 3H, 19-CH_3_), 2.08 (m, 2H, 4-H), 2.16 (s, 3H, 20-CH_3_), 4.01 (q, 2H, -OCH_2_CH_3_, *J* = 7.0 Hz), 5.74 (s, 1H, 14-H), 6.34 (d, 1H, 10-H, *J* = 10.24 Hz); 6.41 (d, 1H, 12-H, *J* = 16.79 Hz), 6.42 (d, 1H, 8-H, *J* = 17.53 Hz), 6.63 (d, 2H, phenyl *J* = 8.82 Hz), 6.67 (d, 1H, 7-H, *J* = 17.53 Hz), 6.83 (dd, 1H, 11-H, *J* = 10.24 Hz, *J* = 16.79), 7.25 (d, 2H, phenyl, J = 8.82 Hz), N–H exchangeable (Appendix A). ^13^C NMR (100 MHz, CD_3_OD): δ 12.80 (C19,20), 14.20 (CH_3_CH_2_O), 19.30 (C3), 20.90 (C18), 28.40 (C16,17), 33.10 (C4), 34.30 (C1), 39.68 (C2), 64.30 (CH_3_CH_2_O), 114.50 (CH = phenyl), 120.52 (CH = phenyl), 125.80 (C11), 126.10 (C14), 127.00 (C7), 129.40 (5), 132.10 (C10), 136.40 (C9), 136.70 (C12), 137.40 (C6), 137.43 (C phenyl), 137.60 (C8), 152.30 (C13), 158.50 (C phenyl), 164.50 (C15, C=O) (Appendix A). Anal. Calcd. for C_28_H_37_NO_2_: C, 80.15%; H, 8.89%; N, 3.34%. Found: C, 80.34%; H, 8.65%; N, 3.05%. GC-MS: *m*/*z* 420.3 (M + 1)^+^, 283.3 (M − 136.1)^+^ (Appendix A).

### 3.3. TLC

The TLC profiles of retinoic acid (Rf = 0.45) and of 4-EPR (Rf = 0.61) prepared according to the procedure developed by us were obtained using aluminum-backed silica gel plates (Merck DC-Alufolien Kieselgel 60 F254, Merck, Washington, DC, USA), and detection of spots was made by UV light (254 nm), using a Handheld UV Lamp, LW/SW, 6W, UVGL-58 (Science Company^®^, Lakewood, CO, USA). The TLCs were eluted in a closed glass developing chamber to keep the environment saturated with solvent vapors using a mixture hexane/acetone 6/4.

### 3.4. HPLC Analyses

The HPLC analysis of 4-EPR was performed on a Hewlet-Packard series 1100 instrument equipped with a RP-18.5 micrometers (LiChro CART 250-4) using a mixture acetonitrile (ACN)/H_2_O/acetic acid (AcOH) 80/18/2 (*v*/*v*/*v*) as eluent (flow rate 1 mL/min, reading at 360 nm, Rt = 20.88 min).

### 3.5. UV–Vis Analyses

The UV–Vis spectra of 4-EPR at concentrations of 0.0048 and 0.0050 mg/mL (MeOH/ACN, 1/1, λ_max_ = 361 nm (ε 52,030) were determined using a UV–Vis instrument (HP 8453, Hewlett Packard, Palo Alto, CA, USA) supplied with a 3 mL cuvette.

### 3.6. FTIR Spectra

FTIR analyses were carried out using a Spectrum Two FT-IR Spectrometer (PerkinElmer, Inc., Waltham, MA, USA). The analyses were made preparing the samples as KBr pellets and the spectra were acquired from 4000 to 600 cm^−1^, with 1 cm^−1^ spectral resolution, co-adding 32 interferograms, with a measurement accuracy in the frequency data at each measured point of 0.01 cm^−1^, due to the internal laser reference of the instrument.

### 3.7. ^1^H and ^13^C NMR Analyses

^1^H and ^13^C NMR spectra were acquired on a Jeol 400 MHz spectrometer (JEOL USA, Inc., Peabody, MA, USA) at 400 and 100 MHz, respectively. Fully decoupled ^13^C NMR spectra were reported. Chemical shifts were reported in ppm (parts per million) units relative to the internal standard tetramethylsilane (TMS = 0.00 ppm), and the splitting patterns were described as follows: s (singlet), d (doublet), t (triplet), q (quartet), m (multiplet), and br was added for broad signals.

### 3.8. GC-MS Analyses

The GC-MS spectrum of 4-EPR was obtained with an Ion Trap Varian Saturn 2000 instrument (Artisan Technology Group ^®^, Champaign, IL, USA), (CI mode, filament current 10 mA) equipped with a DB-5MS (J&W) 30 m, i.d. 0.32 mm, film 1 µm capillary column.

## 4. Conclusions

The absence in the literature of a valid and detailed protocol for the preparation of a compound essential for the determination in tissues of 4-HPR and its metabolites, an operation necessary to allow its preclinical development, supports the relevance of study herein reported. Furthermore, the insignificant availability of 4-EPR on the market further supports the need for disclosing our synthetic method which allows to obtain highly pure 4-EPR. Thanks to the contents of our study, researchers will be able to produce quickly and easily 4-EPR in home.

## Data Availability

All useful data concerning this study are provided in the main text and in the related Appendix A.

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
