# Peer review of "One-Step, Low-Cost, Operator-Friendly, and Scalable Procedure to Synthetize Highly Pure N-(4-ethoxyphenyl)-retinamide in Quantitative Yield without Purification Work-Up"

_molecules, 2022, doi:10.3390/molecules27113632_

Round 1

Reviewer 1 Report

The authors developed in this manuscript a one-step reaction for the synthesis of N-(4-ethoxyphenyl)-retinamide (4-EPR), which is a reference molecule for analysis of 4-HPR (a pre-clinical drug candidate) levels in tissues. The key of the new procedure is the reverse of the equivalents and addition order of the two substrates (ATRA and EAH), which gave the 4-EPR in quantitative yield without column purification. The compound characterization appears to be fine. However, the area of this research is narrow which may not attract the interest of wide readers. Anyway, I would like to recommend this manuscript be published in Molecules after the revision of the followings:

1. Scheme 1: EDCI was lost on the first reaction arrow.

“KHSO4 10%” → “KHSO4 10% aq.”;  “NaOH15%” → “NaOH 15% aq.”

2. Fig 4: What are the two concentrations of the UV-Vis spectra?  

3. Line 170-175: The procedure did not mix the solution of ATRA and EAH.

4. Fig S1 and S2: Integration should be shown in H-NMR. Chemical shifts should be marked in H- and C-NMR.

5. CD3OD was the NMR solvent of HNMR spectra in Fig S1. The solvent peak should be at 3.31 ppm, however, it was shown at around 3.1 ppm.

6. Line 187-189: “1.34 (m, 2H, 2-H), 1.55 (m, 2H, 3-H) … 2.08 (m, 2H, 4-H) … 6.63−7.24 (m, 4H, phenyl)…” The writing style is irregular. For example, the 2H at 2.08 were actually far from each other, one at around 2.18, the other at around 2.02. They should be written separately.

7. In Line 194, there was a peak at 152.3 in CNMR data. However, in Fig S2, there was obviously no peak around 150 ppm.

Author Response

The authors developed in this manuscript a one-step reaction for the synthesis of N-(4-ethoxyphenyl)-retinamide (4-EPR), which is a reference molecule for analysis of 4-HPR (a pre-clinical drug candidate) levels in tissues. The key of the new procedure is the reverse of the equivalents and addition order of the two substrates (ATRA and EAH), which gave the 4-EPR in quantitative yield without column purification. The compound characterization appears to be fine. However, the area of this research is narrow which may not attract the interest of wide readers. Anyway, I would like to recommend this manuscript be published in Molecules after the revision of the followings:

  1. Scheme 1: EDCI was lost on the first reaction arrow.

“KHSO4 10%” → “KHSO4 10% aq.”;  “NaOH15%” → “NaOH 15% aq.”

 We thank the Reviewer for having notified these inaccuracies. The Scheme 1 has been corrected according to the Reviewer’s requests. Please, see Scheme 1, page 2.

  1. Fig 4: What are the two concentrations of the UV-Vis spectra?  

As the Reviewer requested, the two different concentrations of 4-EPR have been added both in the Experimental section (lines 223-224) and in the Figure 4 caption (lines 102-103). 

  1. Line 170-175: The procedure did not mix the solution of ATRA and EAH.

We thank the Reviewer for his comment and apologize for our inaccuracy. The procedure has been corrected (lines 183-185).

  1. Fig S1 and S2: Integration should be shown in H-NMR. Chemical shifts should be marked in H- and C-NMR.

The Reviewer’s request has been satisfied including the chemical shifts in Figure S2 and including the new Figure S1b.

  1. CD3OD was the NMR solvent of HNMR spectra in Fig S1. The solvent peak should be at 3.31 ppm, however, it was shown at around 3.1 ppm.

 Concerning the comment of the Reviewer, we are forced to rebut it. Indeed, the data reported by the Reviewer (3.31 ppm) refers to the chemical shift of the CH3 proton atoms of methanol-d4 in CD3OD (please, see at http://www.nmrs.io/1H/methanol-d4), while the signals present in the spectrum of 4-EPR acquired by us in CD3OD are the signals due to the residual protons in the deuterated methanol. Such signals, as also reported in the Bruker almanac, are two signals at 3.10 ppm (multiplet) and at 4.9 ppm (singlet). Please, see also the attachment (PDF).

  1. Line 187-189: “1.34 (m, 2H, 2-H), 1.55 (m, 2H, 3-H) … 2.08 (m, 2H, 4-H) … 6.63−7.24 (m, 4H, phenyl)…” The writing style is irregular. For example, the 2H at 2.08 were actually far from each other, one at around 2.18, the other at around 2.02. They should be written separately.

Concerning the comment of the Reviewer, we are forced to rebut it. Indeed, for the same example reported by the Reviewer (signals for the two protons named H-4), the two groups of signals (which the Reviewer would want written separate), actually belong to two proton atoms which are magnetically not equivalent, but they are chemically equivalent and consequently they must have the same chemical shift corresponding to the center of the multiplet, i.e. 2.08, as reported. The same for all other signals belonging to the methylene groups of cyclohexene. For confirmation, please see the Supplementary Information associated to the article https://doi.org/10.1016/j.tetlet.2008.12.108 where examples of peaks lists of 1H NMR spectra of cyclohexane derivatives (where the signals of proton atoms of methylene groups are described as multiplets with a single chemical shift), are reported. Anyway, for satisfying point 4, the original 1H NMR spectrum of 4-EPR was carefully checked and the pecks list has been corrected accordingly.

  1. In Line 194, there was a peak at 152.3 in CNMR data. However, in Fig S2, there was obviously no peak around 150 ppm.

We apologize with the Reviewer for our inaccuracy. The Figure S2 was wrong and the correct Figure S2 has been now included in SM. Indeed, to satisfy point 4, the original 13C NMR spectrum of 4-EPR was checked, discovering the incongruence and the correct Figure S2 which agrees with the peaks list in the main text is now present in SM.

Reviewer 2 Report

Title: One-step, Low-Cost, Operator-Friendly, and Scalable Procedure to Synthetize Highly Pure N-(4-ethoxyphenyl)-retinamide in Quantitative Yield Without Purification Work-up

In this research article, the authors have reported a scalable, operator-friendly and one-step procedure to synthetize highly pure 4-EPR [N-(4-ethoxyphenyl)-retinamide] without purification work-up and in quantitative yield. The authors have given a concise introduction about 4-EPR. The authors have characterized the synthesized product using appropriate spectroscopic techniques. The simplicity of this chemistry would make this approach a practical one for adaptation by other research groups. Based on the importance of this work, I suggest publishing in “Molecules” after minor revision. Several suggestions are made for the minor revision.

1)     Scheme 1: Please correct the reaction condition (EDCI reagent is missing in the scheme)

2)     Authors compared the FTIR spectrum of 4-EPR with ATRA and they mentioned that band at 1688 cm-1 belongs to the carboxylic acid group of ATRA. I suggest the authors to give the FTIR spectrum for ATRA along with 4-EPR.

3)     Page 2, line 44: Please correct the name “Chao”. It should be “Cho”

4)     Authors should give the synthetic procedure for ATRA

5)     I suggest the authors to show the previous reported synthetic schemes for 4-EPR in the manuscript.

6)     In the 1H-NMR spectral data of 4-EPR, the total number of protons count is not matching, please check and correct it.

7)     I suggest the authors to show the integration for NMR peaks along with chemical shift values in 1H-NMR and 13C-NMR.

Author Response

Title: One-step, Low-Cost, Operator-Friendly, and Scalable Procedure to Synthetize Highly Pure N-(4-ethoxyphenyl)-retinamide in Quantitative Yield Without Purification Work-up

In this research article, the authors have reported a scalable, operator-friendly and one-step procedure to synthetize highly pure 4-EPR [N-(4-ethoxyphenyl)-retinamide] without purification work-up and in quantitative yield. The authors have given a concise introduction about 4-EPR. The authors have characterized the synthesized product using appropriate spectroscopic techniques. The simplicity of this chemistry would make this approach a practical one for adaptation by other research groups. Based on the importance of this work, I suggest publishing in “Molecules” after minor revision. Several suggestions are made for the minor revision.

  • Scheme 1:Please correct the reaction condition (EDCI reagent is missing in the scheme)

The Scheme has been corrected (page 2).

  • Authors compared the FTIR spectrum of 4-EPR with ATRA and they mentioned that band at 1688 cm-1belongs to the carboxylic acid group of ATRA. I suggest the authors to give the FTIR spectrum for ATRA along with 4-EPR.

As requested, the FTIR spectrum of ATRA has been provided, as Figure 5a. Consequently, the Figure 5 caption has been modified, as well as lines 108-110 in the main text.

  • Page 2, line 44:Please correct the name “Chao”. It should be “Cho”

The name has been corrected.

  • Authors should give the synthetic procedure for ATRA

We make kindly note to the Reviewer that ATRA also known as Tretinoin, is a naturally occurring derivative of vitamin A (retinol) commercially available, since it is not synthetically produced. Consequently, the request of the Reviewer cannot be met.

  • I suggest the authors to show the previous reported synthetic schemes for 4-EPR in the manuscript.

As requested, the previous reported synthetic Schemes have been included in SM as Scheme S1 with the available details. Mentions to Scheme S1 have been included in the main text (lines 47,50 and 53).

6)     In the 1H-NMR spectral data of 4-EPR, the total number of protons count is not matching, please check and correct it.

We apologize with the Reviewer for our inaccuracy. The peaks list has been checked and corrected.

  • I suggest the authors to show the integration for NMR peaks along with chemical shift values in 1H-NMR and 13C-NMR.

The Reviewer’s request has been satisfied including the chemical shifts in Figure S2 and including the new Figure S1b.